# Change of Processes in the COVID-19 Scenario: From Face-to-Face to Remote Teaching-Learning Systems

Cristina Gallego-Gómez [1],*, Carmen De-Pablos-Heredero [2] and José Luis Montes-Botella [3]

1 Department of Marketing, EAE Business School, Joaquin Costa, 41, 28002 Madrid, Spain
2 Department of Business Economics (Administration, Management and Organization), Applied Economics II and Fundamentals of Economic Analysis, Rey Juan Carlos University, Paseo de los Artilleros s/n, 28032 Madrid, Spain; carmen.depablos@urjc.es
3 Department of Applied Economy I, Rey Juan Carlos University, Paseo de los Artilleros s/n, 28032 Madrid, Spain; joseluis.montes@urjc.es
* Correspondence: cristina.gallego@campus.eae.es

**Abstract:** COVID-19 has accelerated digital transformation in teaching-learning environments. Universities based on face-to-face models have had to quickly adapt their processes to ensure the success of remote teaching-learning systems in the last months. The growing demand for technological resources has meant an effort to understand the requirements and variables that affect students' acceptance, intention to use, and adoption of these tools. This study aims to analyze students' acceptance of online processes adopted by universities because of the COVID-19 scenario. Although this study is based on a Technology Acceptance Model (TAM), it also considers other factors, such as perceived efficiency and satisfaction. A questionnaire was built and distributed to 313 students. The data were processed using the Structural Equation Model (SEM) method. The results indicate that 30.7% of the students improved their views of remote education using online systems. However, 49.9% of students do not believe that face-to-face teaching-learning education will be replaced by virtual teaching-learning education in the long term. Our findings confirm that the enriched TAM model built provides a useful theoretical approach to understanding and explaining users' acceptance of remote learning environments when there is a need to rapidly migrate from face-to-face to online teaching-learning processes.

**Keywords:** Technology Acceptance Model (TAM); COVID-19 scenario; Structural Equation Model (SEM); remote learning; change of processes

## 1. Introduction

Technology plays a fundamental role in progress and has led to the redefinition of most products and services. It has enabled new forms of consumption and enabled new ways of learning [1,2].

In this sense [3], digital literacy has been an emerging and growing priority in educational policies and evaluations of governments and institutions from the first years of the 21st century. However, for more than two decades, a debate about the need for a renovation of Spanish universities has been taking place within mass media [4,5]. According to [6], traditional face-to-face education is just one modality of education, but information and communication technologies (ICT) allow educational institutions to offer alternative modalities, such as remote and online education [7–9] or enriching both approaches with hybrid modalities [10]. In this context, the speed of learning is evident due to the fact that schools that use more technology provide learning communities for students to learn digital skills quickly.

Romero, Contreras, and Pérez [11] demonstrated the need to develop transversal actions to train both students and teachers in the field of media competencies so that they can face an ecosystem dominated by fake news and disinformation.

Davis [12], through the Technology Acceptance Model (TAM), found that people's attitudes towards using information technologies were directly related to their perceptions of those technologies. Orlikowski and Gash [13] argued that people's knowledge about different technologies is critical to understanding their interaction with them. TAM models have been applied to best explain those results in higher education [14–17].

The arrival of the COVID-19 pandemic has put the digital model of the university to the test. In this context, remote education is not an option; it is the only way to continue educational processes [18]. Higher education institutions have been forced to implement 100%-remote education systems. As Hodges, Moore, Lockee, Trust, and Bond [19] indicate, there is a clear difference between emergency remote teaching systems and online teaching systems. In particular, the first refers to the systems developed by educational institutions to migrate from face-to-face teaching to remote teaching because of the COVID-19 scenario. Online teaching systems are built around IT possibilities as an alternative to face-to-face, or even remote, teaching systems.

Drašler, Bertoncelj, Korošec, Pajk Žontar, Poklar Ulrih, and Cigi'c [20] affirm that differences in gender attitudes towards online systems can impact final results differently.

This study examines students' acceptance of online processes adopted by universities due to the COVID-19 scenario. Although this study is based on TAM models, it also includes other factors, such as perceived efficiency and satisfaction. This research constitutes a first approach to the situation.

In accordance with previous explanations, this paper analyzes how students perceive the changes that their universities have implemented to adapt to the COVID-19 scenario and how they continue with their normal activities. With our results, universities can implement routines to reinforce the aspects that have been considered positive and work harder to change unsolved or negatively perceived ones.

This research can help universities involved in offering online educational processes understand how students perceive the changes in those processes.

This paper has been divided as follows: Section 1 exposes the theoretical and conceptual background; Section 2 presents the methodology; and Section 3 shows results. Section 4 offers a discussion, and finally, Section 5 offers the conclusions.

### 1.1. Theoretical Framework: Extended Technology Acceptance Model for Analyzing the Adoption of Online Education

The Technology Acceptance Model (TAM), initially developed by Davis [12], is one of the most popular frameworks for analyzing consumer acceptance intentions. TAM analyzes consumer acceptance intentions through perceived usefulness and perceived ease of use [17,21].

The extended TAM [12], which includes the behavioral intention to use technology in the impact on actual system use, has been used to analyze consumer acceptance of education, as shown in Figure 1.

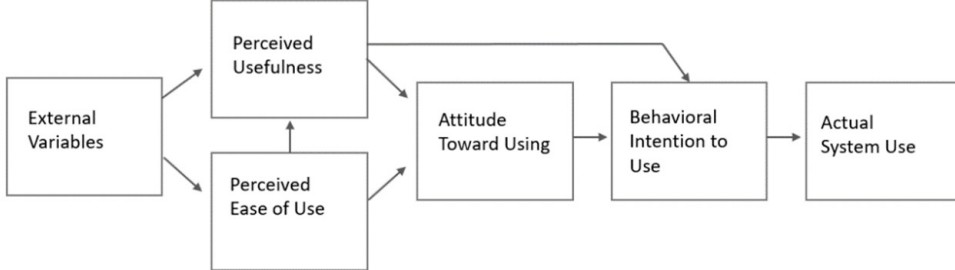

**Figure 1.** TAM Model (Davis, Bagozzi and Warshaw, 1989).

Although previous arguments have been discussed over time [22–24], they still explain users' satisfaction with technology through ease of use, usefulness, and attitude towards using information technology (IT). Concretely, TAM [12] has primarily been used to study

the acceptance of IT concerning innovative education methods [17] and to analyze students' behaviors regarding the use of new technologies and processes [15,25,26].

*1.2. Hypotheses*

The following hypotheses are proposed for the five dimensions: attitude, advantages of use, intent of use, satisfaction, and utility.

### 1.2.1. Attitude

Perceived usefulness is the degree to which a person believes that using a particular system would enhance their job performance [12,22]. Therefore, if students perceive advantages in using online educational resources, their attitude towards them will be more positive.

According to Lin and Lu [27], the perceived usefulness and ease of use of a website are defined as the extent to which the user believes that using the website would increase his/her work performance without too much effort. The attitude, along with subjective norms, determines behavioral intention in education [28]. On this basis, Wu and Chen [29] indicated that the perceived behavioral control (PBC) reflects a person's perception of ease or difficulty towards implementing the behavior out of their interest. These authors state that it would be more comprehensive to understand the behavioral intention of use.

Vijayasarathy [30] finds the attitude and the ease of use being linked to the intention of interest. However, he affirms that this relationship is limited in time and only has an impact at the beginning of the adoption.

Having taken into consideration the previous arguments, the following hypothesis is proposed:

**Hypothesis 1 (H1).** *Positive attitudes towards using online systems would be significantly associated with the intention to use them.*

### 1.2.2. Advantages

One of the most important motivating factors for a student is to obtain easy access to useful information that will help them learn better [31,32]. Following the premises of self-determination theory [33], intrinsic motivation arises from the enjoyment related to participation, while extrinsic motivation relates to social pressures, such as the concern for one's reputation.

Lawler and Porter [33] advocated structuring the work environment so that effective performance would lead to both intrinsic and extrinsic rewards, which would in turn produce total job satisfaction.

Having taken into consideration the previous arguments, the following hypothesis is proposed:

**Hypothesis 2 (H2).** *Students' attitudes towards using online systems depend on students' perceived advantages.*

### 1.2.3. Intent of Use

Empirical results demonstrate that the educational community could do a great deal to enhance students' intention by encouraging gamification strategies, which would lead to the greater application of innovative technological educational tools in face-to-face learning [32]. However, it is not the only factor that helps consolidate the perceived advantages [34,35]. A lecturer's intention to use technology is a determining factor for the student and serves to increases his/her motivation [16]. The same research suggests that one's abilities may be indirectly related to their intentions through the association of those two factors with self-efficacy beliefs.

Moreover, Gudanescu [35] supports building on a constructivist approach to learning; e-learning—as indeed traditional learning—must be perceived as relevant to learners. To

be mastered and retained, content must be connected to things that students already know. Satisfaction with online systems has also been related to the effectiveness of using these systems [36–38]. Having taken into consideration the previous arguments, the following hypothesis is proposed:

**Hypothesis 3 (H3).** *Students' intentions to use online systems depend on the students' perceived advantages.*

### 1.2.4. Utility

Users' willingness to accept technology based on their perceptions can be predicted using the TAM model [14]. Students can also interact with one another during the learning process, which might positively affect adoption. Interaction with instructors and peers can be crucial to learners' satisfaction and can reinforce learners' need to gain competency [39].

In fact, the current trend in education is to incorporate technology in the learning process. Lin and Chen [40] affirm that the learning effect and learning gain are utilized to measure teaching effectiveness. Some students feel frustrated; however, others are motivated [35].

Having taken into consideration the previous arguments, the following hypothesis is proposed:

**Hypothesis 4 (H4).** *Students' perceived satisfaction when using online systems would positively impact the systems' perceived usefulness.*

### 1.2.5. Satisfaction

Online learning's perceived usefulness, ease of use, attitude, and learning behavioral intention have a remarkable positive influence on perceived satisfaction, while perceived usefulness, ease of use, and learning attitude significantly positively affect perceived satisfaction through behavioral intention [41].

Design features and enjoyment only have a significant relationship with students' e-satisfaction without any direct relationship with students' e-retention [42].

Having taken into consideration the previous arguments, the following hypotheses are proposed:

**Hypothesis 5 (H5).** *Students' perceived satisfaction when using online systems will impact student's attitudes towards using them.*

**Hypothesis 6 (H6).** *Students' perceived satisfaction when using online systems will impact their intention to use them.*

## 2. Materials and Methods

### 2.1. Data Collection

To test the hypothesized relationships, we conducted an online survey among people who study at public and private Spanish universities.

To identify problems with the questions, the questionnaire was pre-tested and reviewed by five university lecturers with expertise in both the methodology and subject areas.

The link to the online survey was sent to university lecturers who teach undergraduate courses in different cities of Spain—Madrid, Andalucía, Castilla la Mancha, and Barcelona—to distribute to their students.

Those surveyed were located in the second, third, and fourth grades. They were all students who had had the experience of being enrolled in face-to-face courses during the pandemic.

The investigation was performed using a sample of 313 students from September (26 September 2020) to November (28 November 2020). During this time, the online survey was sent to them.

For this study, public and private universities were sampled. Specifically, the institutions were Carlos the Third University (UC3M), ESIC Business and Marketing School (ESIC), Castilla La Mancha University (CLM), and EAE Business School (Barcelona).

The online questionnaire was carried out through the Google Forms platform, and the link to complete the survey was distributed for two months.

In a first descriptive analysis, different attitudes for women facing men were not found, as previous research predicted [20]. For this reason, in our research, the gender gap in digital use was not considered.

## 2.2. Measures and Method of Analysis

The dependent variable (that is, satisfaction) is measured on a 5-point Likert scale, anchored with totally disagree (1) and totally agree (5). With this, an attempt is made to measure the items that indicate students' willingness to take an online course that uses technology as an intermediary in the future. Respondents were asked if their experience was satisfactory and if, based on it, they might consider adopting non-face-to-face studies.

The following table (Table 1) shows how the variables have been measured and the inspiring authors of each of them.

**Table 1.** Questionnaire, variables, and authors.

| Questionnaire | Variable | Meaning | Inspiring Authors |
|---|---|---|---|
| I consider my self-learning activities to be effective because: (I have an adequate learning environment) | AD_AUTOARE | | |
| I consider my self-learning activities to be effective because: (I communicate and work with my friends to improve my self-learning) | AD_AUTOCOM | | |
| I find my self-learning activities effective because: (I can define my learning goals daily) | AD_AUTOGOA | Advantages of use | [32,43] |
| I believe that self-learning during COVID-19 is necessary because: (I can check my learning progress) | AD_AUTOLEA | | |
| I consider my self-learning activities to be effective because: (I have enough elements for my self-learning) | AD_AUTOSOU | | |
| Regarding the computer or personal computer that you used for your online university courses, | AD_PC | | |
| I consider my self-learning activities to be effective because: (I have good concentration skills) | AT_AUTOCON | | |
| I consider my self-learning activities to be effective because: (I have the support of my family) | AT_AUTOFAM | | |
| I believe that self-learning during COVID-19 is necessary because: (I can maintain my learning habits) | AT_AUTOMAN | | |
| I find my self-learning activities effective because: (I am motivated to self-learn) | AT_AUTOMOT | | |
| During the COVID-19 lockdown, how many hours do you spend for: (Offline learning) | AT_CONHOOFF | | |
| During the confinement for COVID-19, how many hours do you spend for: (Online learning) | AT_CONHOON | Attitude | [27–29,44,45] |
| How many hours do you study a day, on average? (During confinement as a result of COVID-19) | AT_HOURAFT | | |
| How many hours do you study a day on average? (Before confinement as a result of COVID-19) | AT_HOURBEF | | |
| I set my learning and study goals independently in regular university courses taught online. | AT_LEARGOAL | | |
| I organize my studies according to my planning when I take regular university courses dictated online. | AT_LEARPLAN | | |

**Table 1.** *Cont.*

| Questionnaire | Variable | Meaning | Inspiring Authors |
|---|---|---|---|
| I search for learning resources and do my homework independently in regular university courses taught online. | AT_LEARSEAR | | |
| The regular face-to-face courses of the university dictated online . . . (They are an addiction for me) | AT_PONADDI | | |
| The regular face-to-face courses at the university taught online . . . (They make me feel apprehensive or fearful) | AT_PONFEAR | | |
| The regular face-to-face courses of the university dictated online . . . (They have become a habit for me) | AT_PONHABI | | |
| The regular face-to-face courses at the university taught online . . . (They scare me because I can make mistakes that I cannot correct) | AT_PONMISTA | | |
| The regular face-to-face courses of the university dictated online . . . (They have become something natural for me) | AT_PONNORM | | |
| The regular face-to-face courses of the university dictated online . . . (I find them intimidating) | AT_PONPRIV | | |
| I believe that self-learning during COVID-19 is necessary because: (My siblings show me that self-learning is necessary) | IN_AUTOBRO | | |
| I believe that self-learning during COVID-19 is necessary because: (My friends show me that self-learning is necessary) | IN_AUTOFRI | | |
| I believe that self-learning during COVID-19 is necessary because: (My parents show me that self-learning is necessary) | IN_AUTOPAR | | |
| I believe that self-learning during COVID-19 is necessary because: (My teachers show me that self-learning is necessary) | IN_AUTOTEA | | |
| Based on my overall experience of regular university courses taught online: (Many of my expectations were confirmed) | IN_EXPCONFIR | Intended use | [16,32,34,35,46–48] |
| I intend to continue taking regular university courses taught online. | IN_LEARCONTI | | |
| If I could, I would stop taking regular college courses taught online. | IN_LEARNOTONL | | |
| I have high expectations for my learning by taking the regular university courses online. | IN_LEARONL | | |
| I would prefer to continue taking regular university courses taught online, rather than any other option. | IN_LEARONLONLY | | |
| The regular face-to-face courses of the university dictated online . . . (I must continue using them) | IN_PONDO | | |
| How do you evaluate the quality of the internet connection you used for your online university courses? | SA_CONNECTQA | | |
| Based on my overall experience of regular university courses taught online: (I am very displeased with them) | SA_EXPDISL | | |
| Based on my overall experience of regular university courses taught online: (I feel fascinated with them) | SA_EXPFASCI | | |
| Based on my overall experience of regular university courses taught online: (My overall experience was better than I expected) | SA_EXPGLOB | Satisfaction | [36–39,41,42] |
| Based on my overall experience of regular university courses taught online: (The level of services provided was better than I expected) | SA_EXPGOOD | | |
| Based on my overall experience of regular university courses taught online: (I am very satisfied with them) | SA_EXPLIKE | | |
| How do you evaluate your performance in the subjects you took and completed online during the first six months of 2020? | SA_FINONLINE | | |
| The regular face-to-face courses of the university dictated online... (They improve my learning process) | US_PONIMPR | | |
| The regular face-to-face courses of the university dictated online . . . (They improve my academic performance/performance in the learning process) | US_PONPERF | Useful/Utility | [14,32,35,40] |
| The regular face-to-face university courses taught online . . . (It allows me to complete my assignments more easily.) | US_PONTASK | | |
| The regular face-to-face courses of the university dictated online... (They are useful in my learning process) | US_PONUSEF | | |

Once the questionnaire was presented, an explanation of the model applied was presented.

The Structural Equation Model (SEM) is applied in this research. The latent variables (constructs) represent the concepts, and the indicators are the input data. SEM searches for causal relationships between latent variables and assumes complex relationships (with direct and indirect effects) [49].

The method is applied to the data obtained in students' responses to the previously mentioned online questionnaire ($n$ = 313). Data have been analyzed using the Structural Equation Model (SEM) to estimate and test causal links between multiple dependent and independent constructs through a single analysis.

## 3. Results

All 313 individuals were students, 162 of whom (52%) were women and 151 (48%) men. By area of study, 274 belonged to the field of social sciences (economics, business administration, tourism, international relations, and law), 28 belonged to the field of engineering, and 11 to other related fields, such as finance and accounting.

To test the relationships between the indicators and latent constructs and the structural relationships between the latent constructs (Figure 2), we developed a Structural Equation Model (SEM). The model was constructed by applying the Partial Least Squares (PLS) procedure using the Smart PLS 3.3.3. software [49]. The PLS algorithm was chosen according to the following criteria: the relative newness of the phenomenon investigated, its modeling being in an emergent stage, PLS minimal recommendations concerning sample size, prediction accuracy, and comparatively low demands on data multinormality [49–51].

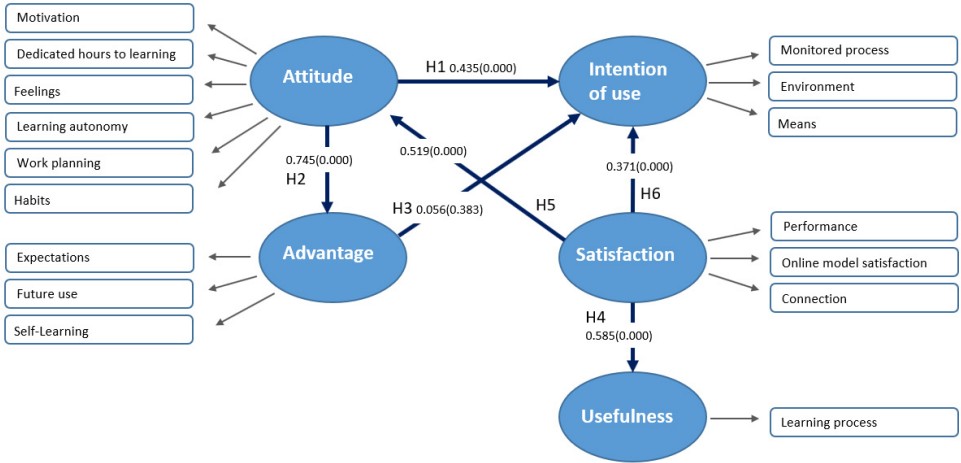

**Figure 2. Figure 2** presents the proposed general model with its constructs and indicators. All the hypotheses proposed in the model are accepted except H3. The numbers on the arrows show the strength of the relationship between constructs measured on a normalized scale from −1 to +1 and (in parentheses) the $p$-value corresponding to the tested hypothesis. Own Elaborated, 2021.

To validate the model, as shown below, we proceeded in two phases: first, we assessed the measurement model; then, we evaluated the structural model. Considering the external (measurement) model, the loadings for the factors are above the 0.600 threshold.

Internal consistency was measured by Cronbach's alpha [52] and by composite reliability (Dillon–Goldstein rho). Both measurements exceeded the minimum proposed values for explorative research by a threshold of 0.600 [44]. Statistical significance was assessed using a resampling bootstrap (Figure 2). As for the indicators' reliability, all the path values are significant ($p$ < 0.01).

Results show a significant positive relationship between attitude and intention of use (H1: β = 0.435, $p$ < 0.000), attitude and advantages (H2: β = 0.745, $p$ < 0.000), satisfaction and usefulness (H4: β = 0.585, $p$ < 0.000), satisfaction and attitude (H5: β = 0.519, $p$ < 0.000), and satisfaction and intention (H6: β = 0.371, $p$ < 0.000). The hypothesis H3, which posed

the relationship between students' intention to use online systems and students' perceived advantages, could not be supported (H3: $\beta = 0.056$, $p < 0373$) at the 95% confidence level.

This research also shows that 30.7% of students have improved their views of online education. However, 49.9% of students do not believe that, in the long term, face-to-face teaching-learning education will be replaced by virtual education.

In view of the results, we can affirm the following:

- RH1: positive attitudes towards using online systems would be significantly associated with the intention to use them.
- RH2: students' attitudes towards using online systems depend on students' perceived advantages.
- RH3: students' intention to use online systems depend on students' perceived advantages.
- RH4: students' perceived satisfaction when using online systems would positively impact the systems' perceived usefulness.
- RH5: students' perceived satisfaction when using online systems will impact students' attitudes towards using them.
- RH6: students' perceived satisfaction when using online systems will impact their intention to use them.

## 4. Discussion

This analysis shows that the enriched TAM model is essential to understanding students' perceptions of how universities have migrated from face-to-face to online teaching environments in the COVID-19 scenario. Positive relationships have been found between attitude and intention of use, attitude and perceived advantages, satisfaction and usefulness, satisfaction and attitude, and satisfaction and intention. The results also evidence how an interesting percentage of students have improved the way they perceive online education now compared with in the past. However, they perceive that the face-to-face education system will be unlikely to be replaced by a virtual system in the long run. According to the results, the measurement model is entirely satisfactory. The reliability of every individual item and the values of the sample and composite reliability are adequate. The independent explanatory variables are satisfactory.

Furthermore, high levels of internal consistency and reliability have been demonstrated among latent variables. The values for validity and the discriminant validity of the measurements are adequate. The hypotheses were checked and validated. The relationships were positive, mostly with high significance, except for the relationship between students' intention to use online systems and students' perceived advantages.

This analysis is aligned with other published research on students' satisfaction with online teaching-learning models. Some of the earlier efforts also applied TAM models [15,16,35,39,45,46]. Others were validated through alternative organizational models, such as relational coordination [8,9,53].

Students have positively evaluated the efforts that universities have made to be adapted to COVID-19 restrictions. However, they do not think online teaching-learning processes will replace face-to-face ones in the long term.

The TAM model has proved robustness in previous analyses. However, it does not specify which types of professional knowledge lecturers must have about teaching and learning with technology to integrate technology meaningfully. Therefore, more attention should be paid to the so-called Technological Pedagogical Content Knowledge (TPACK) framework [16].

The theory of planned behavior (TPB) [12] is the model widely used to discuss these antecedents' effects on behavioral intention. An extension of the Trust and TAM model with TPB would be more comprehensive in providing an understanding of the behavioral intention to use online teaching-learning systems. Furthermore, a large sample survey is used to examine this framework empirically.

## 5. Conclusions

The TAM model constructs were the starting point for the research, and the applicability of TAM as a model for this study is also validated.

The model built confirms a set of indicators that enable analyzing the satisfaction perceived by students adapted to online teaching-learning systems due to the COVID-19 scenario. The study highlights the importance of attitudes, training, and the environment where the teaching-learning process takes place.

Our results confirm that the enriched TAM model built provides a useful theoretical model to help understand and explain users' acceptance of an online learning environment when there is a need to rapidly migrate from face-to-face learning to the online teaching-learning process. Our results also indicate that efficiency, environment, and students' degrees of satisfaction positively influence the original TAM variables as well as students' acceptance of this technology.

This research confirms that positive attitudes of students towards using online systems positively impact their intention to use them. The perceived satisfaction of students when using online systems would also impact the systems' perceived usefulness and the students' attitudes towards using them. Therefore, from a practical perspective, this model can help in the digital transformation process of universities. As attitudes depend on perceptions and training skills, this study contributes to understanding the importance of investing in improving students' attitudes and training in new IT tools and of considering redesigning processes to reach the best results.

As a future topic of investigation, gender inequality in relation to TAM is an important issue. There are authors [20] who affirm differences in female behaviors where more time spent studying did not lead to higher levels of stress among female students

**Author Contributions:** Conceptualization and methodology, all authors. Formal analysis, software, data curation, data processing, J.L.M.-B.; statistical analysis.; validation and investigation, C.G.-G. and C.D.-P.-H.; supervision, J.L.M.-B. and C.D.-P.-H.; data acquisition, C.G.-G. All authors were involved in developing, writing, commenting, editing, and reviewing the manuscript. All authors have read and agreed to the published version of the manuscript.

**Funding:** This research received no external funding.

**Institutional Review Board Statement:** Not applicable.

**Informed Consent Statement:** Not applicable.

**Data Availability Statement:** The data presented in this study are available on request from the corresponding author J.L.-M.B.

**Acknowledgments:** The authors wish to express their gratitude to the students involved as coresearchers who contributed to data collection. Thanks to EAE Business School and Open Innova High Performance Research Group-URJC for their support to this article.

**Conflicts of Interest:** The authors declare no conflict of interest.

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
