# Peer review of "Change of Processes in the COVID-19 Scenario: From Face-to-Face to Remote Teaching-Learning Systems"

_sustainability, doi:10.3390/su131910513_

Round 1
Reviewer 1 Report
The article should not be published in its current state. A number of inconsistencies and contradictions as well as imprecise numerical data (number of sudents = 313 (line 18, 196 and 222), 312 (line 228) 274+28+12=314 (lines 229-231) testify to the need for a revision.
There are some Incomplete sentences make comprehension difficult. e.g., lines 255,256: "The H3 hypothesis, influence of 255 student's perceived advantages on student's, could not be supported..."
Partial verbatim repetition of sentences (e.g., lines 43-45 and lines 75-77).
The numbering of references does not correspond to the order of their appearance in the text.
Author Response
Dear reviewer 1,
we appreciate the effort of the review and your observations, and we communicate that we have taken them into account in the new version of article. Your feedback has allowed us to improve our manuscript. In the following paragraphs you can see the answers to each of your recommendations.
R: The article should not be published in its current state. A number of inconsistencies and contradictions as well as imprecise numerical data (number of sudents = 313 (line 18, 196 and 222), 312 (line 228) 274+28+12=314 (lines 229-231) testify to the need for a major revision.
There are some Incomplete sentences make comprehension difficult. e.g., lines 255,256: "The H3 hypothesis, influence of 255 student's perceived advantages on student's, could not be supported..."
Partial verbatim repetition of sentences (e.g., lines 43-45 and lines 75-77).
A: We have removed the inconsistencies and contradictions as well as the imprecise numerical data throughout the paper as suggested by reviewer. In blue the changes have been stressed in text. The number of students has been corrected in all parts. E.g. the repetition of sentences: previous information repeated in lines 75-77 has been removed. Lines 255-256 (269-271 now) has been rewritten for a better comprehension as follows: “The H3 hypothesis, that posed the relationship between students' intention towards using online systems and student's perceived advantages, could not be supported”.
R: The numbering of references does not correspond to the order of their appearance in the text.
A: The numbering of references has been amended according to Journal’s formatting requirements.
Reviewer 2 Report
You must take in consideration more diferences linked to gender gap in digital use
Author Response
Dear reviewer 2,
we appreciate the effort of the review and your observations, and we communicate that we have taken them into account in the new version of article. Your feedback has allowed us to improve our manuscript. In the following paragraphs you can see the answers to each of your recommendations.
R: You must take in consideration more differences linked to gender gap in digital use.
A: Differences linked to gender gap have been consider in the introduction (lines 57-59): Drašler, Bertoncelj, Korošec, Pajk Žontar, Poklar Ulrih, and Cigi´c [xx] affirm that differences in female attitudes towards online systems can impact in final results differently facing males.
In data collection, we explain the reasons why we do not consider the gender gap in our analysis (lines 212-214): In a first descriptive analysis, different attitudes of females facing men are not found as previous research described (Drašler et al., 2021). For this reason, in our re-search the gender gap in digital use has not been considered.
We also include the gender gap as an interesting future area of research in this topic (lines 344-346): As a future line, gender inequality in relation to TAM is an important issue. There are authors [20] who affirm differences in female behaviors where more time spent studying did not lead to higher levels of stress among female students.
Reviewer 3 Report
Interesting manuscript focusing of the acceptance of online learning in Spanish universities. Please see the following comments for revisions and recommended changes:
The notion of optimization mentioned in the title is not present in the main text and should be removed.
All in-text citations need to be revised as they must be numbered in order of appearance in the text. Hence, the first two references (L30) will be [1,2] and not [4,46].
L49-50: The pandemic triggered world-wide emergency remote teaching, a modality with distinct differences from online learning [1]. Please make a clear distinction between the two concepts.
L292: Add an appropriate reference to the Theory of planned behavior.
L300-312: The confirmation of hypotheses should be moved to the Results section. Also, in Figure 2 explain the obtained relative results and their comparative strength.
L324-5: This is a significant statement as it associated with the contribution of the manuscript that should be better justified and explained.
References
- Hodges, C.; Moore, S.; Lockee, B.; Trust, T.; Bond, A. The difference between emergency remote teaching and online learning. Educ. Rev. 2020, 27.
Author Response
Dear reviewer 3,
we appreciate the effort of the review and your observations, and we communicate that we have taken them into account in the new version of article. Your feedback has allowed us to improve our manuscript. In the following paragraphs you can see the answers to each of your recommendations.
R: Interesting manuscript focusing of the acceptance of online learning in Spanish universities. Please see the following comments for revisions and recommended changes:
R: The notion of optimization mentioned in the title is not present in the main text and should be removed
A: We have changed the notion of optimization of processes in title by change of processes that is in agreement with the notion used in text.
R: All in-text citations need to be revised as they must be numbered in order of appearance in the text. Hence, the first two references (L30) will be [1,2] and not [4,46].
A: All in-text citations have been revised and numbered in in order of appearance in the text.
R: L49-50: The pandemic triggered world-wide emergency remote teaching, a modality with distinct differences from online learning [1]. Please make a clear distinction between the two concepts.
References
- Hodges, C.; Moore, S.; Lockee, B.; Trust, T.; Bond, A. The difference between emergency remote teaching and online learning. Educ. Rev. 2020, 27.
A: The distinction between the two concepts has been included according to the suggested reference (lines 51-56): As Hodges, Moore, Lockee, Trust, and Bond [x] indicate, there is a clear difference be-tween emergency remote teaching systems and on line teaching systems. First ones refer to the systems developed by education Institutions because of COVID-19 scenario to migrate face-to-face teaching style to remote teaching. Online teaching systems are teaching styles built around IT possibilities as an alternative to face-to-face, even remote teaching systems.
L292: Add an appropriate reference to the Theory of planned behavior
A: The appropriate reference to the Theory of planned behavior has been added (L297): Theory of planned behavior (TPB) [Davis, 1989 XX]
L300-312: The confirmation of hypotheses should be moved to the Results section. Also, in Figure 2 explain the obtained relative results and their comparative strength
A: The confirmation of hypothesis has been moved to results section (L 266-278)
In view of the results, we can affirm that:
- RH1: positive attitudes towards using online systems would be significantly associated with the intention to use them.
- RH2: Students' attitude towards using online systems depends on student's perceived advantages.
- RH3: Students' intention towards using online systems depends on student's perceived advantages.
- RH4: students' perceived satisfaction when using online systems would positively impact their perceived usefulness.
- RH5: students' perceived satisfaction when using online systems will impact student's attitude towards using them.
- RH6: students' perceived satisfaction when using online systems will impact their intention to use them.
We add: Figure 2 presents the proposed general model with its constructs and indicators. All the hypotheses proposed in the model are accepted except H3. The numbers on the arrows show: the strength of the relationship between constructs measured on a normalized scale from -1 to +1 and (in parentheses) the p-value corresponding to the tested hypothesis
R.L324-5: This is a significant statement as it associated with the contribution of the manuscript that should be better justified and explained.
A: The statement has been justified and explained (L339-345): “This research confirms that positive attitudes of students towards using on line systems impacts positively in their intention of using them. The perceived satisfaction of students when using on line systems would also impact on their perceived usefulness and towards their attitude towards using them. Therefore, from the practical perspective, this model can help in the digital transformation process of universities. As attitudes depend on perceptions and training skills, it contributes to understanding the importance of investing in improving students' attitudes and training in new IT tools and considering redesigning processes to reach the best results. “
Reviewer 4 Report
Dear Authors,
There are many things you have to take into consideration before publishing:
- English language is awful
- You are not consistent through the paper, e.g. COVID-19 should be written like this, you write it in different forms.
- There are not enough keywords
- You do not clearly explain the models you use. Perhaps the readers do not know what they are about.
- You should make your paper more scientific.
Author Response
Dear reviewer 4,
we appreciate the effort of the review and your observations, and we communicate that we have taken them into account in the new version of article. Your feedback has allowed us to improve our manuscript. In the following paragraphs you can see the answers to each of your recommendations.
R: English language is awful
A: We have revised and corrected the English language.
R: You are not consistent through the paper, e.g. COVID-19 should be written like this, you write it in different forms.
A: COVID-19 has been written as suggested by reviewer in all the text
R: There are not enough keywords
A: We have added more key words.
R: You do not clearly explain the models you use. Perhaps the readers do not know what they are about.
A: We have explained the models used in the research.
R: You should make your paper more scientific.
A: The changes done in the new version of the text make the paper more scientific.
Round 2
Reviewer 1 Report
The references are now sorted correctly. However, inconsistencies still exist: If the sum of participants is n=313, how can 162 women and 152 men have participated? Likewise, 274+28+12=313? (lines 241-244)
The abbreviation TAM is not explained in the abstract, for abbreviation SEM there are several explanations: M stands for "model" on the one hand and for "modeling" on the other hand. In principle, abbreviations should be explained before their first appearance in the text and then used consistently.
Note for further improvement: table 1 breaks up the reading flow in section 2.2, but is not mentioned anywhere in the text. For better readability, it would be worth considering whether its content could be better presented as a separate section, structured according to the 5 viewpoints (column "Meaning" in table 1).In doing so, the inspiring authors could be referenced accordingly. The names of the variables seem to be of less interest to readers, especially since they are not mentioned anywhere in the text.
Author Response
Dear reviewer 1,
We appreciate the effort of the review and your observations. We have taken them into account in the new version of article. Your feedback has allowed us to improve our manuscript. In the following paragraphs you can see the answers to each of your recommendations.
R: The references are now sorted correctly. However, inconsistencies still exist: If the sum of participants is n=313, how can 162 women and 152 men have participated? Likewise, 274+28+12=313? (lines 241-244)
A. The men are 151 and the other´s students 11. We have corrected the mistakes.
R: The abbreviation TAM is not explained in the abstract, for abbreviation SEM there are several explanations: M stands for "model" on the one hand and for "modeling" on the other hand. In principle, abbreviations should be explained before their first appearance in the text and then used consistently.
A: TAM has been explained in the abstract
A: SEM explained as Structural Equation Model. It was replaced in the text-
R: Note for further improvement: table 1 breaks up the reading flow in section 2.2, but is not mentioned anywhere in the text. For better readability, it would be worth considering whether its content could be better presented as a separate section, structured according to the 5 viewpoints (column "Meaning" in table 1). In doing so, the inspiring authors could be referenced accordingly. The names of the variables seem to be of less interest to readers, especially since they are not mentioned anywhere in the text.
A: We have presented the explanation to table 1 in text. We have decided to maintain the name of the variables too in case readers are interested in replicating the analysis that we have done. We understand that, for this purpose, the table 1, such as it is presented, may be of help.
So, we indicate in text:
“The following table 1 (table 1) shows how the variables have been measured and the inspiring authors for each of them”.
Reviewer 3 Report
Authors addressed all identified issues in a satisfactory way and improved significantly the quality of the mansucript. They need to pay attention to and rectify some minor proofing issues, e.g. L155 (too many pararaphs, lack of dot), L342 (italics).
Author Response
Dear reviewer 3,
We appreciate the effort of the review and your observations. We have taken them into account in the new version of article. Your feedback has allowed us to improve our manuscript. In the following paragraphs you can see the answers to each of your recommendations.
R: Authors addressed all identified issues in a satisfactory way and improved significantly the quality of the mansucript. They need to pay attention to and rectify some minor proofing issues, e.g. L155 (too many pararaphs, lack of dot), L342 (italics).
A: Now there is only one paragraph. Dots have been included and italics removed
Besides, Gudanescu [42] confirms that the building off a constructivist approach to learning, e-learning—as indeed traditional learning—must be perceived as relevant to learners. To be mastered and retained, content must be connected to things they already know. Satisfaction with online systems has also been related to effectiveness in using these systems [43-46]. Having into consideration previous arguments, the following hypothesis is proposed:
Reviewer 4 Report
Dear Authors,
Thank you for taking into account the changes. The paper look better now and is more logically structured.
Author Response
Dear reviewer 4,
We appreciate the effort of the review and your observations. We have taken them into account in the new version of article. Your feedback has allowed us to improve our manuscript. In the following paragraphs you can see the answers to each of your recommendations.
R: Dear Authors,
Thank you for taking into account the changes. The paper looks better now and is more logically structured.
A: Thank you so much